# The influence of travel time to health facilities on stillbirths: A geospatial case-control analysis of facility-based data in Gombe, Nigeria

Oghenebrume Wariri[1,2]*, Egwu Onuwabuchi[2,3], Jacob Albin Korem Alhassan[2,4], Eseoghene Dase[2,5], Iliya Jalo[6], Christopher Hassan Laima[3], Halima Usman Farouk[3], Aliyu U. El-Nafaty[3‡], Uduak Okomo[1‡], Winfred Dotse-Gborgbortsi[7,8‡]

1 Vaccines and Immunity Theme, MRC Unit The Gambia at the London School of Hygiene and Tropical Medicine, Fajara, The Gambia, 2 African Population and Health Policy Initiative, Gombe, Gombe State, Nigeria, 3 Department of Obstetrics and Gynaecology, Federal Teaching Hospital Gombe, Gombe, Nigeria, 4 Department of Community Health and Epidemiology, College of Medicine, University of Saskatchewan, Saskatoon, Canada, 5 Department of Obstetrics and Gynaecology, Cedarcrest Hospital, Abuja, Nigeria, 6 Department of Paediatrics, Federal Teaching Hospital Gombe, Gombe, Nigeria, 7 School of Geography and Environmental Science, University of Southampton, Southampton, United Kingdom, 8 WorldPop Research Group, School of Geography and Environmental Science, University of Southampton, Southampton, United Kingdom

‡ These authors are joint senior authors on this work.
* Oghenebrume.Wariri@lshtm.ac.uk

**Data Availability Statement:** All relevant data are within the manuscript and its Supporting information files.

## Abstract

Access to quality emergency obstetric and newborn care (EmONC); having a skilled attendant at birth (SBA); adequate antenatal care; and efficient referral systems are considered the most effective interventions in preventing stillbirths. We determined the influence of travel time from mother's area of residence to a tertiary health facility where women sought care on the likelihood of delivering a stillbirth. We carried out a prospective matched case-control study between 1st January 2019 and 31st December 2019 at the Federal Teaching Hospital Gombe (FTHG), Nigeria. All women who experienced a stillbirth after hospital admission during the study period were included as cases while controls were consecutive age-matched (ratio 1:1) women who experienced a live birth. We modelled travel time to health facilities. To determine how travel time to the nearest health facility and the FTHG were predictive of the likelihood of stillbirths, we fitted a conditional logistic regression model. A total of 318 women, including 159 who had stillborn babies (cases) and 159 age-matched women who had live births (controls) were included. We did not observe any significant difference in the mean travel time to the nearest government health facility for women who had experienced a stillbirth compared to those who had a live birth [9.3 mins (SD 7.3, 11.2) vs 6.9 mins (SD 5.1, 8.7) respectively, p = 0.077]. However, women who experienced a stillbirth had twice the mean travel time of women who had a live birth (26.3 vs 14.5 mins) when measured from their area of residence to the FTHG where deliveries occurred. Women who lived farther than 60 minutes were 12 times more likely of having a stillborn [OR = 12 (1.8, 24.3), p = 0.011] compared to those who lived within 15 minutes travel time to

**Funding:** The author(s) received no specific funding for this work.

**Competing interests:** The authors have declared that no competing interests exist.

the FTHG. We have shown for the first time, the influence of travel time to a major tertiary referral health facility on the occurrence of stillbirths in an urban city in, northeast Nigeria.

## Background

An estimated 2.6 million stillbirths are recorded globally every year, with the majority dispro-portionately occurring in low-and middle-income countries (LMIC), particularly in sub-Saha-ran Africa [1]. The World Health Organization (WHO) defines stillbirth as any third trimester foetal death (≥ 28 weeks' gestation) or death of a newborn during childbirth [2]. While the average stillbirth rate (SBR) in many high-income countries range between 2–5 per 1000 births, the rates reported in LMICs are more than ten-fold higher [1]. Nigeria ranked second among the top ten high stillbirth burden countries and had the second highest SBR globally in 2015 [1]. In 2014, the World Health Assembly-backed *Every Newborn Action Plan* (ENAP) acknowledged the need to reduce stillbirths occurring in LMICs thereby setting a global target to reduce SBR to 12 or fewer stillbirths per 1000 births in all countries by 2030 [3]. To achieve this ambitious ENAP stillbirth target by 2030, evidence-based and data-driven policy interven-tions targeted at the individual level, the broader health system and socioeconomic disadvan-tages which determine overall health in the first instance must be prioritised.

The timing of a stillbirth reflects different aspects of maternal health care. An antepartum stillbirth (APSB), is said to have occurred when a baby dies in the mother's womb before the onset of labour (typically more than 12 hours before delivery) and reflects the quality of ante-natal care accessible before and during pregnancy [4, 5]. An intrapartum stillbirth (IPSB) is defined as fetal death during labour, or within 12 hours before delivery, and reflects the quality of obstetric and newborn care available to a pregnant woman during labour, birth and imme-diately after birth [4, 5]. About half of stillbirths are intrapartum, and the majority are consid-ered preventable [1]. Research and interventions targeted at maternal and fetal medical causes of stillbirths have received relatively more attention. However, there is evidence to suggest that limited physical accessibility to quality obstetric and newborn care and adverse socioeconomic determinants of health contribute to stillbirths [6].

Access to quality emergency obstetric and newborn care (EmONC); having a skilled atten-dant at birth (SBA); adequate antenatal care; and efficient referral systems are considered the most effective interventions to prevent adverse pregnancy outcomes [7]. However, in many LMICs, access to SBA or EmONC services can be difficult due to several delays, especially for populations living in rural settings or urban slums where access to roads and means of trans-portation are suboptimal, rarely available, or unaffordable [8–10].

The *'three delays'* conceptual framework developed by Thaddeus and Maine identifies three groups of factors or delays which may limit access to maternal health services leading to mor-tality [11]. These challenges are the delay in deciding to seek care, delay in arrival at a well-equipped health facility and the delay in receiving adequate care. The existing literature sug-gests that difficulties in geographic accessibility to facilities which provide maternal health ser-vices may indirectly contribute to the first-level delay by creating a lack of interest to seek care, and directly impact the second-level delay by increasing the travel time in reaching health facil-ities which provide life-saving services [12–14]. The delay in deciding and reaching a well-equipped health facility on time increases the likelihood of maternal and newborn complica-tions such as stillbirths. Also, they reflect the broader health system performance and

responsiveness [6, 9, 15]. Therefore, IPSB is a sensitive marker of delays in the rapid delivery of a compromised fetus and low-quality care during childbirth.

Despite the substantial estimated burden of stillbirths in Nigeria, there is persistent inaccurate data on its predictors to inform appropriate policy interventions. Most studies focus on reporting stillbirth prevalence rates and exploring their clinical determinants [16–19]. Also, available studies investigated geographic accessibility factors known to strongly influence the use of skilled birth attendant and EmONC for pregnant women but not stillbirth outcome [20, 21]. Furthermore, existing local stillbirth literature shows substantial regional disparity as they are mainly focused on southern Nigeria [16–19].

To bridge the important evidence gap on the relationship between geographic accessibility and stillbirths, we set out to determine the influence of travel time from mother's home to the nearest government health facility and to the major tertiary health facility where they sought care on the likelihood of delivering a stillbirth.

## Methods

### Study setting

This study was conducted at the Federal Teaching Hospital, Gombe (FTHG), a major tertiary health facility located in Gombe City, the capital of Gombe State, northeast Nigeria. Gombe State shares borders with five other states, namely Adamawa, Bauchi, Borno, Taraba, and Yobe. Gombe State is predominantly rural, occupies a total land area of about 20,265sqkm, has an estimated population of 2.9 million people, a population density of 148 per $km^2$, and an annual population growth rate of 4.05% [22, 23]. Most women access maternity services through state-funded public-sector primary and secondary health facilities. Gombe State has more than 600 public-sector and private health facilities spread across 11 Local Government Areas (the equivalent of a district) [24]. More than 90% of the health facilities in Gombe State are primary-level facilities offering basic preventative/curative care, while only about 4% are secondary and tertiary-level facilities offering specialised care [24]. Fig 1 shows the study area map.

The FTHG is the only tertiary hospital in Gombe (see Fig 1 for location). It has 450-bed capacity that offers specialised care, funded by the Federal (central) government, and receives referrals from Gombe and surrounding States. The hospital provides Comprehensive Emergency Obstetric and Newborn Care (CEmONC) and adequately staffed with obstetricians, gynaecologists, midwives, anaesthetists as well as neonatologists. They perform safe blood transfusion, caesarean sections, assisted vaginal delivery, and resuscitation of the newborn. The hospital records averagely 2,400 deliveries annually, with 27% of all births delivered by caesarean section. Between 2010 and 2018, the FTHG annual SBR ranged from 42 per 1000 births (95% CI:34,51) to 65 per 1000 births (95% CI: 55,76), with 52% of all stillbirths being intrapartum [26].

### Study design

We carried out a prospective case-control study at the Obstetrics department of the FTHG between 1st January 2019 and 31st December 2019. All women who experienced a stillbirth after hospital admission during the study period were included as cases while controls were consecutive age-matched women who experienced a live birth. The case to control ratio was 1:1 (i.e. individual matching), and age-matched controls were within two years standard deviation of their respective cases. Women whose pregnancies culminated in multiple births were excluded. Although we did not have a predetermined sample size, we consider that our sample can be representative of a larger population as we included all stillbirths (considered rare

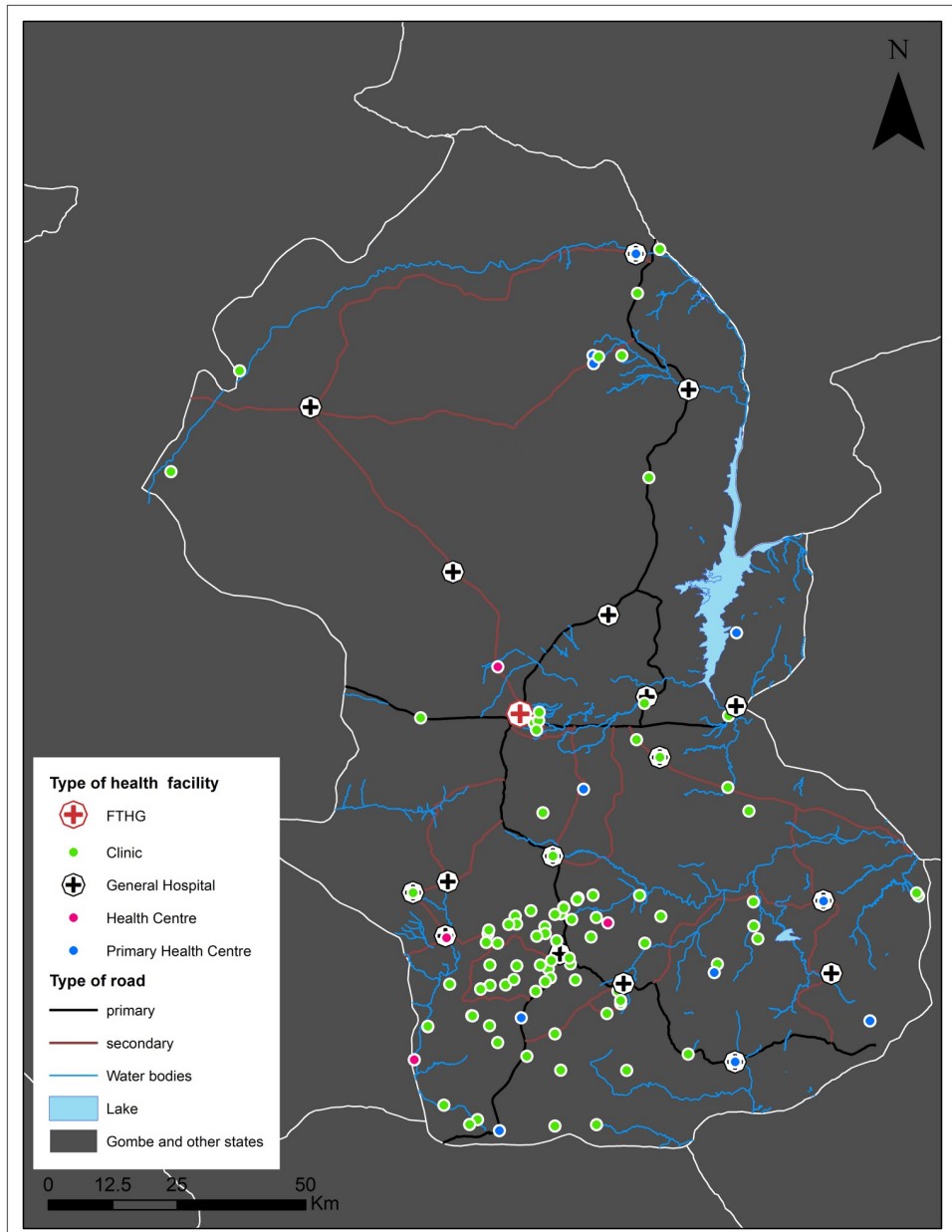

**Fig 1. Study area showing roads, water bodies, and health facilities included in estimating travel times.** Note: This map was produced by the authors with administrative boundaries data from geoBoundaries [25].

events) that occurred over a year period with their appropriate controls in our study setting–a major referral facility.

## Data collection

Written informed consent was obtained from all eligible women after delivery before a pre-tested, researcher administered questionnaire (S1 File) was used to collect information. The questionnaire was developed, pre-tested and adapted based on stillbirth data from our study setting [26]. In our study setting teenage marriage is common, thus, we considered the

participants who were below the age of 18 years (the traditional age of consenting in our setting) as 'emancipated minors' because all of them were already married [27]. Informed consent (rather than assent) was thus sought from these participants in a similar manner to those women who were 18 years or older. Data collected includes their obstetric history, social, economic, and demographic characteristics. Also, their mode of transport to the hospital and referral pathway before arriving at the FTHG for delivery were collected. Cases and their respective controls were approached with the study information after delivery and informed consent for participation in the study was sought. All participants were informed that participation in the study was voluntary and a decision not to participate will not impact the care they will normally receive post-delivery. There was a recruitment window lasting from the day of delivery until seven days afterwards to allow for some recovery from the stress of a stillbirth.

We geocoded the town address of participants using their house address and a smartphone to enable spatial analysis. All addresses were geocoded to the town level for confidentiality and privacy purposes. Due to the larger size and population density of Gombe City, we geocoded suburbs, generally at one square kilometre spatial resolution as towns and used their centroids. All other locations where women came from were geocoded as towns using OpenStreetMap [26]. Therefore, the suburbs of Gombe City and the other locations had relatively similar sizes.

For the location of health facilities, we included the geocoordinates of all government-run public health facilities near the residential areas of participants. The coordinates were obtained from an open-source spatial database of health facilities managed by the public health sector in sub-Saharan Africa curated by the WHO [28].

## Modelling travel time to health facilities

Travel time to health facilities was modelled in AccessMod5.0 [29]. Travel time was chosen to model physical geographic access because it is a better measure that incorporates elevation, road network, and travel speed among other factors that influence geographic accessibility compared to network and straight-line distances [30]. Furthermore, we used AccessMod5.0 because it is free software, simple to use and widely used for analysing geographic accessibility to health services [21, 31]. Travel times were modelled to two destinations, first to the nearest government health facility (i.e., public primary and secondary facilities, excluding dispensaries) then to FTHG (the major referral facility in Gombe) where all the cases and controls delivered their babies. To estimate travel times, we used land cover [32], roads and rivers [33], digital elevation model [34], and the location of health facilities [28]. The travel speed used to estimate travel times varied by road (primary = 100kmh$^{-1}$, secondary = 50kmh$^{-1}$, tertiary = 30kmh$^{-1}$) and land cover type adapted from previous studies [31, 35]. We assumed 10 kmh$^{-1}$ on tracks for motorbikes, tricycles and other types of improvised ambulances used to transport women in labour. Details of the travel speeds applied by landcover and type of road are included as Table in S1 Table.

To avoid creating artificial bridges across water bodies, road segments that intersect water bodies but not fully crossing it due to digitising, conversion or other topological error were corrected using the "clean artefacts" option in AccessMod [29]. The clean artefact function removes only the artificial bridge and includes the other segments of the road in the model. The estimated travel times account for variations in walking and bicycling speed due to changing elevation when travelling towards a health facility. The corrections for walking speed due to changing elevation was implemented with the Tobler's formula while bicycling speed was adjusted using a complex physical model based on velocity, power and resistance that are explained into details in the AccessMod user manual [29]. Finally, we extracted the average travel times within a kilometre distance of the woman's residential town. Then, we calculated

the extra time travelled using the difference between travel time to the nearest health facility and the FTHG.

## Statistical analysis

All statistical analyses were performed with Statistical Package for the Social Sciences (SPSS) (IBM, NY, version 24), figures were generated using ggplot2 in R and maps were created using ArcGIS® software (version 10.4) by Esri [36]. Freely available to use state outline data from geoBoundaries [25], and OpenStreetMap [33] basemaps were used to draw the map figures. We produced two maps, one showing stillbirth or live births layered on travel times and the second showing flows of women towards FTGH.

Summary tables for maternal sociodemographic and geographic accessibility characteristics were generated, firstly for cases (APSB and IPSV) versus controls (live births) and then for cases alone. Categorical and continuous variables were summarised as proportions and means respectively. Cross-tabulations comparing cases versus controls and IPSB versus APSB were performed. Independent sample t-test was used to compare means between groups and chi-square/Fischer's exact test for association between groups, with statistical significance defined as alpha less than 0.05 (two-sided).

We fitted a conditional logistic regression model to predict the likelihood of stillbirths. The independent variables in the regression model were travelling time to the nearest health facility (at intervals of 5 mins), and FTHG (at intervals of 15 mins). The crude regression model was adjusted for known confounders, including the level of education, maternal occupation, parity, booking status, and mode of transport to the hospital on the day of delivery. The confounders were selected *a priori* based on the literature on predictors of stillbirths in sub-Saharan Africa. We report adjusted odds (AOR) ratios and 95% confidence interval (CI).

## Ethics

This study was reviewed and approved by the Research and Ethics Committee (REC) of the Federal Teaching Hospital Gombe (NHREC/25/10/2013). Informed consent was sought from all study participants before participation in this study.

## Results

### Characteristics of cases and controls

A total of 318 women, including 159 who had stillborn babies (cases) and 159 age-matched women who had live births (controls) were included, as shown in Table 1. Their ages ranged from 15 to 50 years, with an average woman 28.6 (SD: 6.6) years old. There was no significant difference in age between the cases and controls (p = 0.217). Women who experienced a still-birth and those who had live births differed significantly with regard to their education, parity, booking status, referral status, mode of transport to the FTHG on day of delivery, occupation, and fathers' occupation (p<0.001 across all indicators). Compared to those who had live born babies, a higher proportion (44.6% vs 13.8%) of women who had experienced a stillbirth lacked formal education; had four or more children (48.1 vs 30.2%), and had been referred (73.0% vs 17.0%) to the FTHG from another facility. Conversely, a higher proportion (26.4% vs 8.1%) of women who had experienced a live birth were formally employed, had husbands who were for-mally employed (61% vs 30.6%) and had their pregnancies booked at the FTHG (67.9% vs 17.5%).

Table 2 compares the characteristics of participants by APSB and IPSB. Although the women who had experienced an IPSB were relatively three years older APSB, the age

**Table 1. Summary characteristics of all women who had stillbirths and their age-matched control who delivered live babies at the Federal Teaching Hospital Gombe from 1st January to 31st December 2019.**

| | All births | | | |
|---|---|---|---|---|
| | **Stillbirths (159) n (%)** | **Live births (159) n (%)** | **Total (318)** | **p value** |
| **Mean age (SD)** | 28.5 (6.8) | 28.6 (6.3) | 28.6 (6.6) | 0.217 |
| **Mother's education** | | | | |
| No formal education | 72 (44.6) | 22 (13.8) | 94 (29.8) | |
| Primary education | 22 (13.8) | 9 (5.7) | 31 (9.7) | |
| Secondary education | 38 (23.8) | 65 (40.9) | 103 (32.3) | |
| Tertiary education | 27 (16.9) | 63 (39.6) | 90 (28.2) | <0.001 |
| **Mother's occupation** | | | | |
| Unemployed | 99 (62.5) | 95 (59.7) | 194 (61.1) | |
| Informal employment | 47 (29.4) | 22 (13.8) | 69 (21.6) | |
| Formal employment | 13 (8.1) | 42 (26.4) | 55 (17.2) | <0.001 |
| **Father's occupation** | | | | |
| Unemployed | 1 (0.6) | 0 (0.0) | 1 (0.3) | |
| Informal employment | 109 (68.8) | 61 (38.4) | 170 (53.6) | |
| Formal employment | 49 (30.6) | 98 (61.6) | 147 (46.1) | <0.001 |
| **Parity** | | | | |
| Nulliparous | 12 (7.5) | 5 (3.1) | 17 (5.3) | |
| 1 to 3 children | 71 (44.4) | 106 (66.7) | 177 (55.5) | |
| 4 or more children | 76 (48.1) | 48 (30.2) | 124 (39.2) | <0.001 |
| **Booking status** | | | | |
| Unbooked | 35 (21.9) | 22 (13.8) | 57 (17.9) | |
| Booked elsewhere | 96 (60.6) | 29 (18.2) | 125 (39.5) | |
| Booked in FTHG | 28 (17.5) | 108 (67.9) | 136 (42.6) | <0.001 |
| **Referral** | | | | |
| No | 43 (27.0) | 132 (83.0) | 175 (54.9) | |
| Yes | 116 (73.0) | 27 (17.0) | 143 (45.1) | <0.001 |
| **Facility referred from** | | | | |
| Primary health centre | 40 (35.0) | 9 (33.3) | 49 (34.7) | |
| Secondary health facility | 56 (47.9) | 12 (44.4) | 68 (47.3) | |
| Tertiary health facility | 8 (6.8) | 2 (7.4) | 10 (6.9) | |
| Private health facility | 12 (10.3) | 4 (14.8) | 16 (11.1) | 0.122 |
| **Transport FTHG for delivery** | | | | |
| Ambulance | 6 (3.8) | 2 (1.3) | 8 (2.5) | |
| Commercial vehicle | 97 (61.3) | 64 (40.3) | 161 (50.8) | |
| Motorcycle | 8 (5.0) | 2 (1.3) | 10 (3.1) | |
| Personal vehicle | 48 (30.0) | 91 (57.2) | 139 (43.6) | <0.001 |

**Note**: FTHG = Federal Teaching Hospital Gombe, SD = Standard Deviation.

difference was not statistically significant (p = 0.134). In contrast to the observed differences between women who had experienced a stillbirth or live birth in Table 1, we did not observe any significant differences between women who had IPSB and those with APSB across all the maternal indicators.

**Table 2. Summary characteristics of all women who had stillbirths delivered at the Federal Teaching Hospital Gombe from 1st January to 31st December 2019 by type of stillbirth.**

| | Stillbirths | | | |
|---|---|---|---|---|
| | Intrapartum stillbirth (96) n (%) | Antepartum stillbirth (63) n (%) | Total (159) | p value |
| **Mean age (SD)** | 30.0 (7.0) | 27.4 (6.0) | 28.5 (6.8) | 0.134 |
| **Mother's education** | | | | |
| No formal education | 43 (45.8) | 29 (46.0) | 72 (45.6) | |
| Primary education | 15 (14.6) | 7 (11.1) | 22 (13.8) | |
| Secondary education | 22 (22.9) | 16 (25.4) | 38 (23.8) | |
| Tertiary education | 16 (16.7) | 11 (17.5) | 27 (16.9) | 0.344 |
| **Mother's occupation** | | | | |
| Unemployed | 60 (63.5) | 39 (61.9) | 99 (62.5) | |
| Informal employment | 29 (29.2) | 18 (28.6) | 47 (29.4) | |
| Formal employment | 7 (7.3) | 6 (9.5) | 13 (8.1) | 0.614 |
| **Father's occupation** | | | | |
| Unemployed | 0 (0.0) | 1 (1.6) | 1 (0.6) | |
| Informal employment | 66 (69.8) | 43 (68.3) | 109 (68.8) | |
| Formal employment | 30 (30.2) | 19 (30.2) | 49 (30.6) | 0.593 |
| **Parity** | | | | |
| Nulliparous | 8 (8.3) | 4 (6.3) | 12 (7.5) | |
| 1 to 3 children | 39 (40.6) | 32 (50.8) | 71 (44.4) | |
| 4 or more children | 49 (51.0) | 27 (42.9) | 76 (48.1) | 0.609 |
| **Booking status** | | | | |
| Unbooked | 21 (21.9) | 14 (22.2) | 35 (21.9) | |
| Booked elsewhere | 61 (63.5) | 35 (55.6) | 96 (60.6) | |
| Booked in FTHG | 14 (14.6) | 14 (22.2) | 28 (17.5) | 0.676 |
| **Referral** | | | | |
| No | 23 (22.9) | 20 (31.7) | 43 (26.9) | |
| Yes | 73 (77.1) | 43 (68.3) | 116 (73.1) | 0.125 |
| **Facility referred from** | | | | |
| Primary health centre | 25 (35.1) | 15 (34.9) | 40 (35.0) | |
| Secondary health facility | 40 (54.1) | 17 (39.5) | 57 (48.7) | |
| Tertiary health facility | 4 (5.4) | 4 (9.3) | 8 (6.8) | |
| Private health facility | 4 (5.4) | 7 (16.3) | 11 (9.4) | 0.357 |
| **Transport FTHG for delivery** | | | | |
| Ambulance | 4 (4.2) | 2 (3.3) | 6 (3.8) | |
| Commercial vehicle | 55 (57.3) | 42 (66.7) | 97 (61.3) | |
| Motorcycle | 4 (4.2) | 4 (6.3) | 8 (5.0) | |
| Personal vehicle | 33 (34.4) | 15 (23.8) | 48 (30.0) | 0.324 |

**Note**: FTHG = Federal Teaching Hospital Gombe, SD = Standard Deviation.

## Travel time and pregnancy outcomes

Travel times for cases and controls to the nearest health facility and FTHG are shown in Table 3. Overall, we did not observe any significant difference in the mean travel time to the nearest government health facility for women who had experienced a stillbirth compared to those who had a live birth [9.3 mins (SD 7.3, 11.2) vs 6.9 mins (SD 5.1, 8.7) respectively, p = 0.077]. When measured from their residential town to the FTHG, women who experienced a stillbirth had approximately twice (26.3 vs 14.5 mins) the mean travel time of women who

**Table 3. Travel time (in minutes) to the nearest health facility and to Federal Teaching Hospital Gombe from participants residential areas compared across stillbirths versus live births, and fresh versus macerated stillbirths among patient seen from 1st January to 31st December 2019.**

| | All cases | | | | Stillbirths | | | |
|---|---|---|---|---|---|---|---|---|
| | Stillbirths (159) | Live Births (159) | Total | p value | Intrapartum stillbirth | Antepartum stillbirth | Total | p value |
| Time to nearest facility (mins); mean (95%CI) | 9.3 (7.3, 11.2) | 6.9 (5.1, 8.7) | 8.1 (6.8, 9.4) | 0.077 | 9.6 (7.2, 12.1) | 8.9 (5.4, 12.3) | 9.3 (7.3, 11.2) | 0.718 |
| Time to FTHG (mins); mean (95%CI) | 26.3 (20.6, 32.0) | 14.5 (10.8, 18.3) | 20.4 (16.9, 23.9) | 0.001 | 27.6 (20.8, 34.4) | 24.2 (13.8, 34.6) | 26.3 (20.6, 32.0) | 0.571 |
| Extra time travelled to FTHG (mins); mean (95%CI) | 17.0 (12.3, 21.4) | 7.6 (5.2, 10.1) | 12.3 (9.7, 14.9) | <0.001 | 17.9 (12.2, 23.7) | 15.3 (8.0, 22.6) | 17.0 (12.3, 21.4) | 0.561 |

**Note**: FTHG = Federal Teaching Hospital Gombe

Extra time travelled = Travel time to FTHG minus travel time to the nearest facility from participant's residential area

Reported p-values for comparing means are based on independent sample t-test and chi-square/Fischer's exact test for association between groups.

had a live birth (Table 3). Fig 2 shows the comparative travel time to health facilities by pregnancy outcome and type of stillbirth. The proportion of women in all categories decreased with increasing distance. More of the women who experienced a live birth lived within five minutes (82.4% vs 63.1%, p = 0.002) travel time to their nearest government health facility

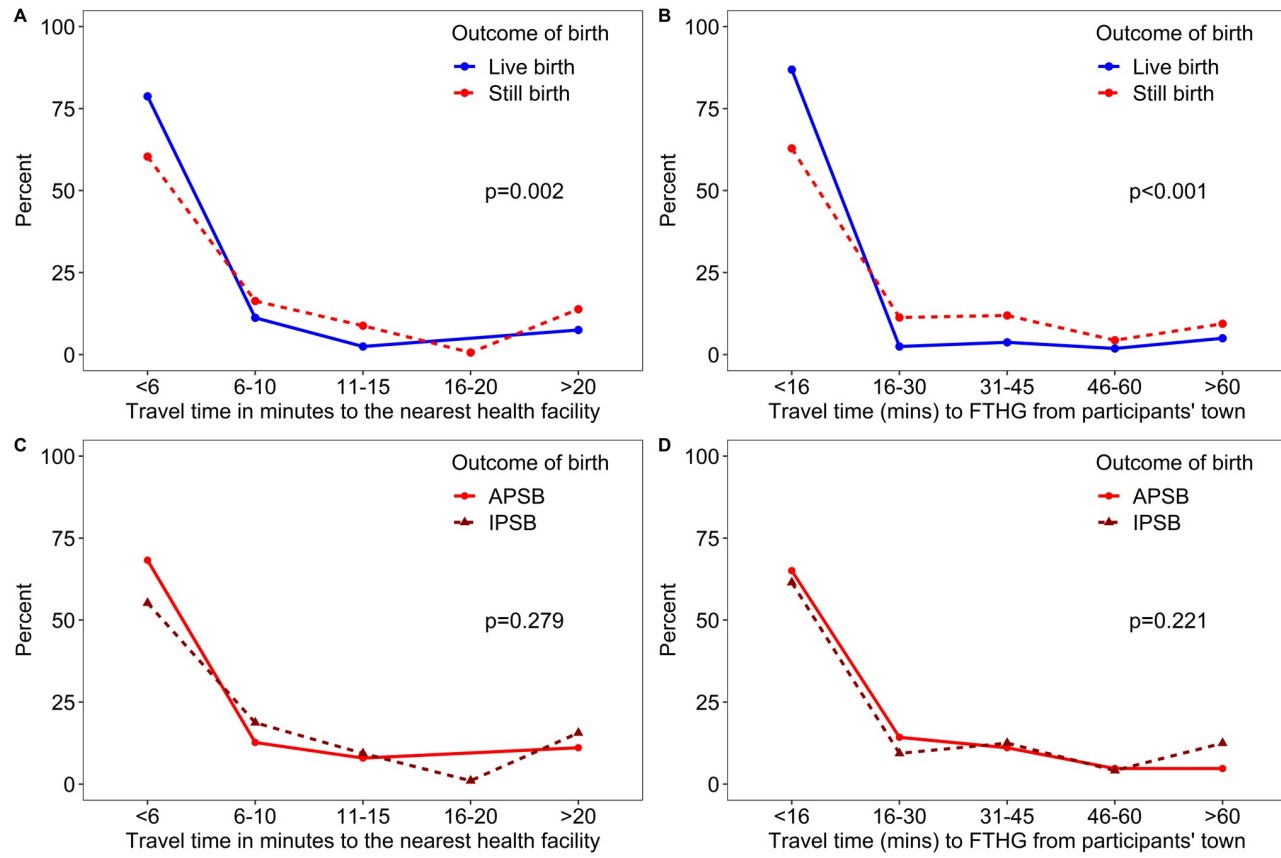

**Fig 2. Comparative distance decay for cases (stillbirths) versus controls (live births) and for women with intrapartum versus those with Antepartum stillbirth from 1st January to 31st December 2019 at the Federal Teaching Hospital Gombe.**

compared to women who had experienced a stillbirth. Similarly, a significantly higher proportion (88.0% vs 62.5%, p<0.001) of women who delivered a live baby lived within 15 minutes travel time to the FTHG compared to those who were delivered of stillborn babies (Fig 2).

The difference in travel time between APSB and IPSB to the nearest health facility and FTHG were not statistically significant. Similarly, the difference in the extra time travelled from the nearest health facility to FTHG between APSB and IPSB was not significant. There was no association between distance groups to the nearest health facility and FTHG by stillbirth cases (Fig 2C and 2D). Meanwhile, the distance groups were associated with cases and controls.

After adjusting for known confounders (mother's age, education, occupation, parity, booking status, referral, and mode of transport to FTHG), there was no difference in the effect of varying travel times to the nearest health facility on the likelihood of experiencing a stillbirth (Table 4). Travel time from residential town to the FTHG differed significantly between the women who had experienced a stillbirth compared to those who had a live birth and predicted the likelihood of stillbirths (Fig 3). The patterns of travel from towns to the FTHG shows many of the stillbirth cases lived within Gombe City close to the hospital (Fig 4). There were 11 cases that travelled almost 30 km to reach FTHG. Also, the flow patterns show communities with a high number (at least five) of stillbirths from outside Gombe city.

The odds of experiencing a stillbirth increased significantly with increasing travel time to FTHG. Women within 30 minutes travel were more than twice as likely to experience a stillbirth as those living within 15 minutes or less travel time, and this increased to almost nine times for women residing beyond one hour's travel time. After adjusting for known confounders, the key predictor of experiencing a stillbirth was a travel time to the FTHG of 60 minutes and above. Women who lived farther than 60 minutes had a 12 times likelihood of having a stillborn (OR = 12, 95% CI = 1.8–24.3, p = 0.011) compared to those who lived within a 15 minutes travel time to the FTHG (Table 4).

**Table 4. Conditional logistic regression model table of the effect of travel time on stillbirths (compared to live births) among babies delivered at the Federal Teaching Hospital Gombe from 1st January to 31st December 2019.**

| | Unadjusted | | | Adjusted* | | |
|---|---|---|---|---|---|---|
| | Odds Ratios | 95% CI | p value | Odds Ratios | 95% CI | p value |
| **Time to nearest facility categories** | | | | | | |
| 5 mins and below | 1 | Reference | 1 | 1 | Reference | 1 |
| 6–10 mins | 2.4 | 1.1, 5.0 | 0.024 | 1.8 | 0.6, 5.5 | 0.277 |
| 11–15 mins | 6.6 | 1.7, 21.6 | 0.006 | 5.1 | 0.9, 28.0 | 0.059 |
| 16–20 mins | 1.3 | 0.08, 20.9 | 0.855 | 0.1 | 0.0, 5.2 | 0.275 |
| 21 mins and above | 2.3 | 1.1, 5.0 | 0.024 | 0.7 | 0.3, 2.0 | 0.504 |
| **Time to FTHG categories** | | | | | | |
| 15 mins and below | 1 | Reference | 1 | 1 | Reference | 1 |
| 16–30 mins | 2.6 | 1.1, 6.4 | 0.036 | 0.9 | 0.3, 3.4 | 0.947 |
| 31–45 mins | 4.7 | 1.1, 23.9 | 0.049 | 0.5 | 0.1, 3.5 | 0.485 |
| 46–60 mins | 4.4 | 1.7, 11.4 | 0.002 | 1.2 | 0.3, 4.2 | 0.822 |
| 60 mins and above | 8.8 | 2.5, 30.6 | 0.001 | 12.2 | 1.8, 24.3 | 0.011 |

*Adjusted for mother's education, occupation, parity, booking status, referral, and mode of transport to FTHG on the day of delivery.

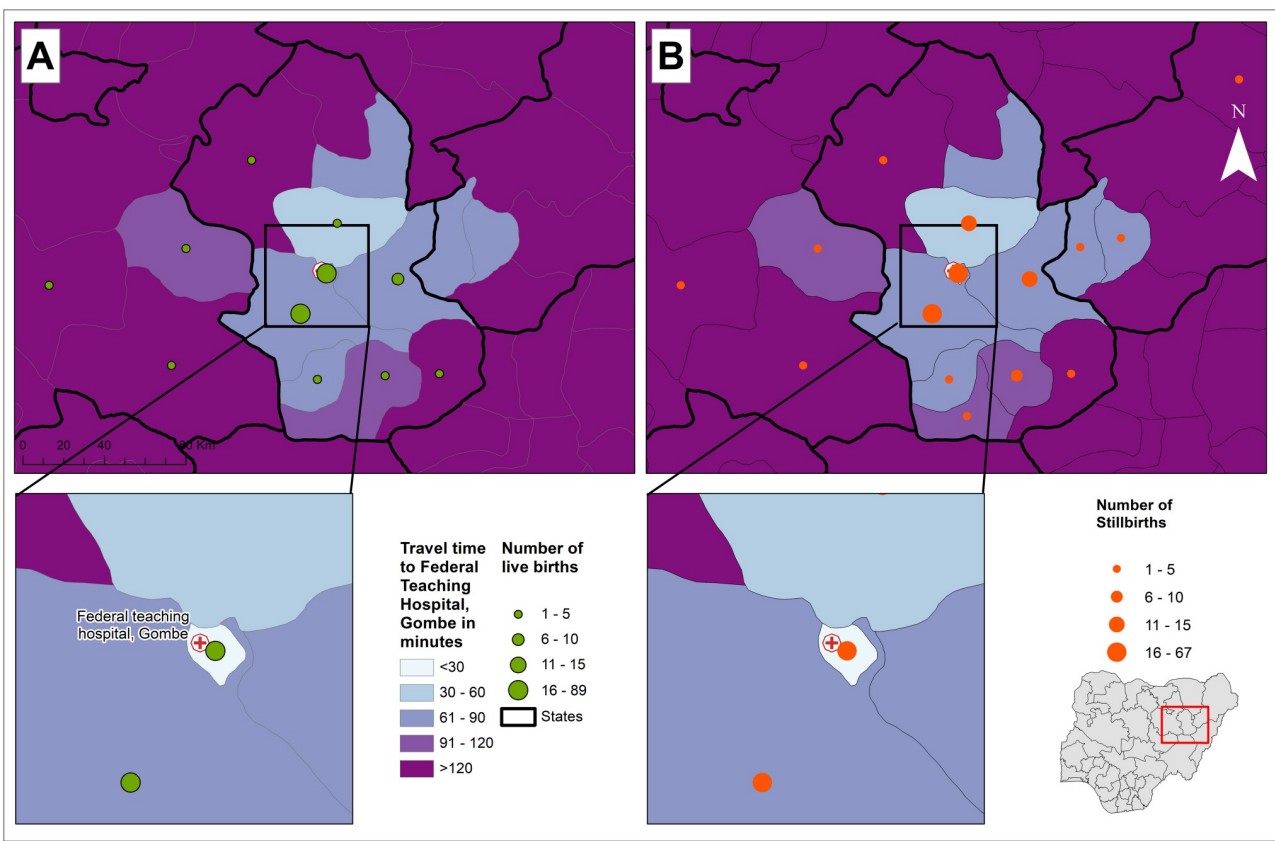

**Fig 3. Travel time from participants' residential area to Federal Teaching Hospital Gombe for; [A] women with Stillbirths and [B] age-matched controls who had Live births from 1st January to 31st December 2019.** *Note*: *This map was produced by the authors with administrative boundaries data from geoBoundaries* [25], *and Base map and data from OpenStreetMap and OpenStreetMap Foundation* [33].

## Discussion

In this study, we explored the influence of travel time to the health facility on the occurrence of stillbirths in Gombe City, northeast Nigeria. We observed a strong association between travel time and pregnancy outcome among women who delivered at the main tertiary referral hospital, FTHG. Travel time to the FTHG among women who experienced stillbirths was twice that of women who experienced a live birth. After adjusting for known confounders, women who lived beyond one-hour travel time to FTHG had a 12-fold significantly higher likelihood of experiencing a stillbirth compared to women living within a quarter of an hour's travel time.

All the women in this study, irrespective of pregnancy outcome, could have received care within eight minutes travel because 90% of government primary and secondary health facilities in Gombe State provide basic obstetric care [24]. However, they travelled averagely 12 extra minutes to reach FTHG, with stillbirth cases travelling even longer than live births. Although FTHG is the major tertiary referral hospital in Gombe state, it is unlikely that all the women in the study were referred there. Hence, it would have been useful to ascertain reasons for the choice of delivery at FTHG among women who were not referred there and had a nearer health facility. The quality of maternal and newborn services offered in public-sector primary and secondary facilities in Nigeria is variable and mostly considered suboptimal due to the differences in resources dedicated to the health system by subnational governments [37]. The

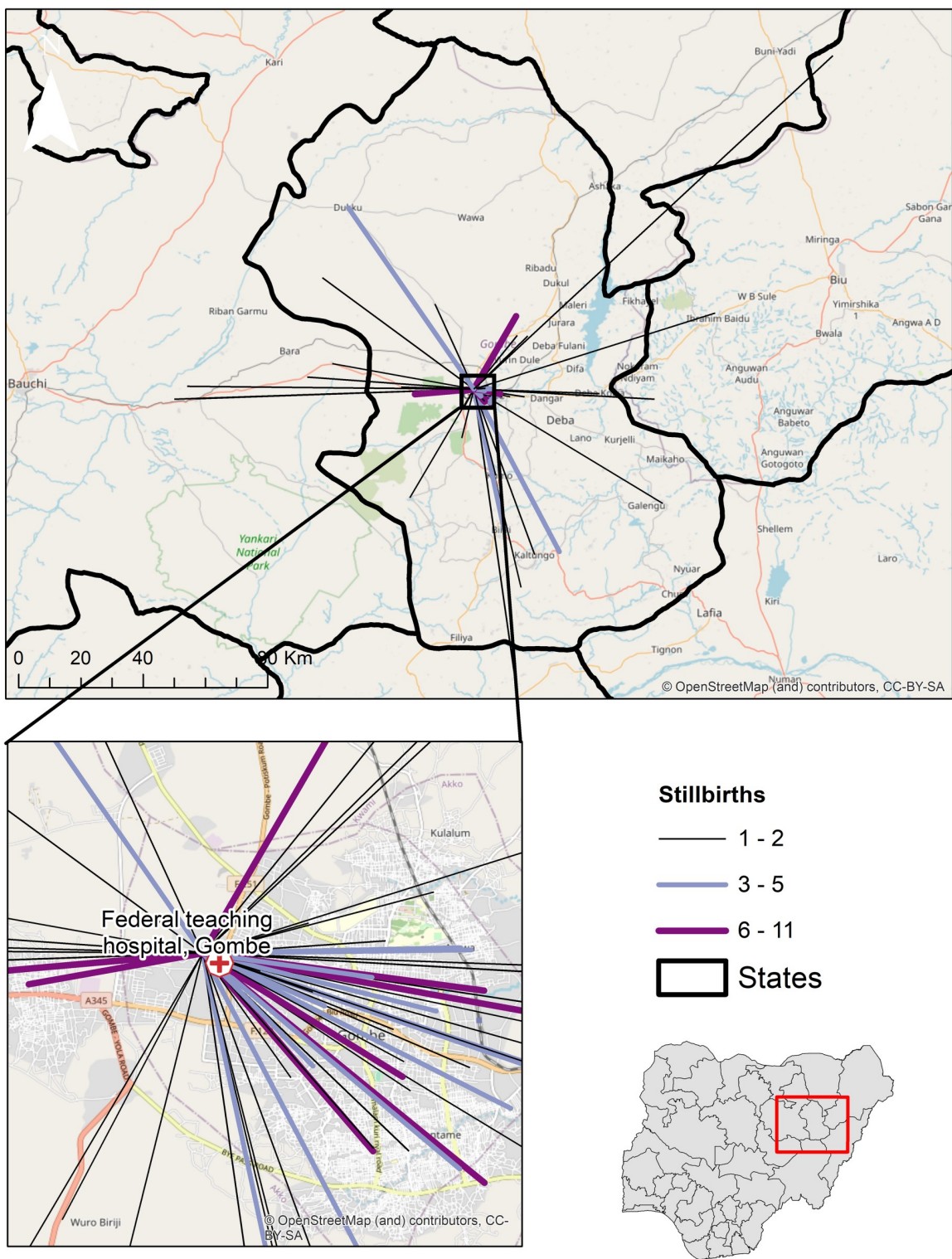

**Fig 4. Travel flows from towns to the Federal Teaching Hospital Gombe.** The increasing width and colour of lines show the number of stillbirths from towns. **Note**: *This map was produced by the authors with administrative boundaries data from geoBoundaries* [25], *and Base map and data from OpenStreetMap and OpenStreetMap Foundation* [33].

available resources and poorer quality could partly explain why many women in Gombe spent extra time travelling to seek care at the distant tertiary FTHG.

This study shows, for the first, that women who delivered stillbirths in Gombe State had a significantly higher travel time than those with live born newborns, similar to a previous study in the Netherlands [38]. Likewise, another study that pooled data from 21 low-and middle-income countries found that adverse perinatal outcomes were associated with significantly longer travel times/distances to healthcare facilities [39]. Clearly, understanding the influence of travel time on the delivery of stillbirths is complex, given its implicit associations with broader individual socioeconomic characteristics such as poverty, level of education, and obstetric characteristics such as parity and booking status [40, 41]. In emphasis, our study also revealed that women who had stillbirths were comparatively poorer, less educated, more likely to be multiparous and unbooked. Women without formal education in our setting are more likely to belong to the poorest socioeconomic group with consequent pre-pregnancy anaemia; have poor knowledge about the warning signs of stillbirth; be undernourished, and miss ante-natal care appointments [26]. Continued exposure to adverse and overlapping socioeconomic determinants of health is known to limit the probability of women to surmount additional hindrances, such as geographic access to delivery services. Consequently, they are at greater risk of poorer health outcomes including stillbirths as previously shown in Gombe [42], Nepal [43], and Spain [44] where stillbirths increased significantly among poorer and less educated women.

The actual travel time to our facility may have been impacted by additional barriers such as ownership of personal vehicle and time spent during referrals between facilities. Our analysis showed that women with stillbirths were more likely to have been referred from other facilities and travelled to the FTHG on day of delivery with non-personal vehicle compared to those with live births, as has been reported in other studies [11, 13]. While we adjusted for referral status and mode of transportation to our facility, we were unable to capture the additional time the women spent in the referral facility, time spent travelling between facilities and time to arrange her transport. These additional times would have likely increased the travel time for women who had stillbirths since a significantly higher proportion of them were referred and travelled on non-personal vehicles compared to those who delivered live babies. We consider that these pre-hospital delays would be longer for women that reside more than an hour away from the FTHG and likely rural dwellers. Furthermore, women living closer (within a quarter of an hour) will reside predominantly in the urban metropolis of Gombe and have a higher likelihood of seeking delivery care directly at the FTHG, instead of elsewhere before being referred. These nuanced differences could have also contributed to the high likelihood of stillbirths for women residing farther away from the FTHG. Future studies could investigate the referral patterns to FTHG, the quality of EmONC services at lower-tier health facilities and their impact on pregnancy outcomes.

Overall, our findings on the influence of travel time on stillbirths, taken together with previous studies on stillbirths in Gombe [26, 41], adds a new perspective to understanding why stillbirths rates continue to be high in our setting. Our findings could inform plans for ensuring and centralising quality emergency obstetric and newborn services to bridge the inequality gap in access to and use of maternal health services. Besides, socioeconomic disadvantages which have contributed to stillbirths in Gombe State needs addressing. Furthermore, interventions should include training and re-training of the local health workforce; reorganisation of services provided by public primary and secondary facilities; and provision of functional government ambulance system which have improved the quality of EmONC elsewhere [43]. Tackling the constraints of geographic access for women with stillbirths identified from this study requires improvements in social infrastructure, which is outside the direct purview of the routine

healthcare system. Although challenging, especially in a resource-limited setting like the north-east of Nigeria where Gombe is located, all levels of government must make evidence-informed policy decision and show leadership at implementing interventions to remedy this situation. Ending preventable stillbirths is achievable only through an integrated approach that addresses the broader socioeconomic, and geographic factors and not just isolated vertical programmes which address clinical causes alone [45].

There are several advantages to the spatial methods used in this study. The journey origin and health facility used were clearly defined, and geographic coordinates were available to measure proximity compared with limitations in similar studies [46]. We avoided the assumption in DHS survey data analysis that women use the health facility closest to their residence [12], as there is evidence of large variations in expected (nearest health facility) and observed patterns of using birthing services in health facilities [47]. Our geospatial analysis also eliminated errors associated with self-reported distance and recall bias in other studies [37].

However, our study has several limitations to be considered in interpreting results. Firstly, we used the town of residence instead of the actual home address in modelling travel time. While generalising the address to town level for all women in a locality protects the confidentiality and privacy of the study participants, it may have led to acceptable aggregation error [48]. A second limitation is that we assumed a uniform speed and travel mode for all women, but there could be variations as not all women will use mechanised transport on roads. Lastly, our travel time modelling did not account for diurnal and seasonal variations in road traffic conditions due to flash flooding, religious festivals and other events which results in traffic congestion prevalent in urban settings.

## Conclusion

We have shown for the first time, the influence of travel time to a major tertiary referral health facility on the occurrence of stillbirths in an urban city in, northeast Nigeria. We did not observe any significant difference in the mean travel time to the nearest government health facility among study participants. However, women who experienced stillbirths had twice the mean travel time from their residence to the facility where they sought care compared to those with live births. There was a positive relationship between longer travel time ($\geq$60 minutes) and delivery of stillbirths. Our finding could guide interventions aimed at ensuring the availability of quality obstetric and newborn care to reduce the unacceptably high stillbirth rates in our setting.

## Supporting information

**S1 File. Pretested questionnaire (data collection tool).**
(DOCX)

**S2 File. Manuscript dataset.**
(XLSX)

**S1 Table. Landcover, road class and their corresponding speeds for modelling travel times.**
(DOCX)

## Acknowledgments

The authors wish to acknowledge the labour ward manager, Balkisu Mamman and staff of Obstetrics and Gynaecology Department of FTHG for providing care to the study participants.

## Author Contributions

**Conceptualization:** Oghenebrume Wariri, Egwu Onuwabuchi, Jacob Albin Korem Alhassan, Eseoghene Dase, Winfred Dotse-Gborgbortsi.

**Data curation:** Oghenebrume Wariri, Egwu Onuwabuchi, Eseoghene Dase, Christopher Hassan Laima, Halima Usman Farouk, Aliyu U. El-Nafaty.

**Formal analysis:** Oghenebrume Wariri, Jacob Albin Korem Alhassan, Winfred Dotse-Gborgbortsi.

**Methodology:** Oghenebrume Wariri, Egwu Onuwabuchi, Jacob Albin Korem Alhassan, Eseoghene Dase, Iliya Jalo, Aliyu U. El-Nafaty, Uduak Okomo, Winfred Dotse-Gborgbortsi.

**Project administration:** Oghenebrume Wariri, Egwu Onuwabuchi, Christopher Hassan Laima, Halima Usman Farouk.

**Resources:** Oghenebrume Wariri.

**Supervision:** Iliya Jalo, Aliyu U. El-Nafaty, Uduak Okomo, Winfred Dotse-Gborgbortsi.

**Visualization:** Winfred Dotse-Gborgbortsi.

**Writing – original draft:** Oghenebrume Wariri, Uduak Okomo.

**Writing – review & editing:** Oghenebrume Wariri, Egwu Onuwabuchi, Jacob Albin Korem Alhassan, Eseoghene Dase, Iliya Jalo, Christopher Hassan Laima, Halima Usman Farouk, Aliyu U. El-Nafaty, Uduak Okomo, Winfred Dotse-Gborgbortsi.

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
