## [Decision Letter · Decision Letter 0]

29 Oct 2020

PONE-D-20-30030

The influence of travel time to health facilities on stillbirths: a geospatial
case-control analysis of facility-based data in Gombe, Nigeria

PLOS ONE

Dear Dr. Wariri,

Thank you for submitting your manuscript to PLOS ONE. After careful consideration, we
feel that it has merit but does not fully meet PLOS ONE’s publication criteria as it
currently stands. Therefore, we invite you to submit a revised version of the
manuscript that addresses the points raised during the review process.

Please submit your revised manuscript by Dec 13 2020 11:59PM. If you will need more
time than this to complete your revisions, please reply to this message or contact
the journal office at plosone@plos.org. When
you're ready to submit your revision, log on to https://www.editorialmanager.com/pone/ and select the 'Submissions
Needing Revision' folder to locate your manuscript file.

If you would like to make changes to your financial disclosure, please include your
updated statement in your cover letter. Guidelines for resubmitting your figure
files are available below the reviewer comments at the end of this letter.

We look forward to receiving your revised manuscript.

Kind regards,

Frank T. Spradley

Academic Editor

PLOS ONE

2. Please include additional information regarding the survey or questionnaire used
in the study and ensure that you have provided sufficient details that others could
replicate the analyses. For instance, if you developed a questionnaire as part of
this study and it is not under a copyright more restrictive than CC-BY, please
include a copy, in both the original language and English, as Supporting
Information. Moreover, please include more details on how the questionnaire was
pre-tested, and whether it was validated.

3. In your Methods section, please provide additional information about the
participant recruitment method and the demographic details of your participants.
Please ensure you have provided sufficient details to replicate the analyses such
as: a) the recruitment date range (month and year), b) a description of any
inclusion/exclusion criteria that were applied to participant recruitment, c) a
table of relevant demographic details, d) a statement as to whether your sample can
be considered representative of a larger population, e) a description of how
participants were recruited, and f) descriptions of where participants were
recruited and where the research took place.

4. We suggest you thoroughly copyedit your manuscript for language usage, spelling,
and grammar. If you do not know anyone who can help you do this, you may wish to
consider employing a professional scientific editing service.  

6. We note that Figure 2 in your submission contain map images which may be
copyrighted. All PLOS content is published under the Creative Commons Attribution
License (CC BY 4.0), which means that the manuscript, images, and Supporting
Information files will be freely available online, and any third party is permitted
to access, download, copy, distribute, and use these materials in any way, even
commercially, with proper attribution. For these reasons, we cannot publish
previously copyrighted maps or satellite images created using proprietary data, such
as Google software (Google Maps, Street View, and Earth). For more information, see
our copyright guidelines: http://journals.plos.org/plosone/s/licenses-and-copyright.

(a) You may seek permission from the original copyright holder of Figure(s) [#] to
publish the content specifically under the CC BY 4.0 license.

(b) If you are unable to obtain permission from the original copyright holder to
publish these figures under the CC BY 4.0 license or if the copyright holder’s
requirements are incompatible with the CC BY 4.0 license, please either i) remove
the figure or ii) supply a replacement figure that complies with the CC BY 4.0
license. Please check copyright information on all replacement figures and update
the figure caption with source information. If applicable, please specify in the
figure caption text when a figure is similar but not identical to the original image
and is therefore for illustrative purposes only.

Reviewers' comments:

Reviewer's Responses to Questions

**Comments to the Author**

1. Is the manuscript technically sound, and do the data support the conclusions?

Reviewer #1: Yes

2. Has the statistical analysis been performed
appropriately and rigorously? 

Reviewer #1: Yes

3. Have the authors made all data underlying the
findings in their manuscript fully available?

Reviewer #1: Yes

4. Is the manuscript presented in an intelligible
fashion and written in standard English?

Reviewer #1: Yes

5. Review Comments to the Author

Reviewer #1: Wariri et al look at the influence of travel time to health facilities
on stillbirths in Gombe, Nigeria. This is an exciting study in attempt at shedding
light on the myriad of factors that contribute to the unacceptably high rate of
still births in most African countries. However, they need to address the issues
below to improve the quality of the publication.

1. What is the influence of existing referral strategy on decision making on where to
seek care first by mothers? Is there a bypassing dynamic in the study population?
What shapes it?

2. It would have been nice to show flow of patients to the tertiary hospital in a map
to better understand the influence of residence on choice of facility.

3. Geographically inaccessibility directly affects the second delay in the three
delay concept by Thaddeus and Maine, rather than the indirect impact it has on the
first delay. Discussion on second delay needed as well.

4. Is there a recommended travel time to nearest hospital facilities which reduces
occurrence of stillbirths during delivery?

5. Include the precise coordinate location of the study setting, inclusion of study
area map will be helpful.

6. Why did you select the travel time model via AccessMod? Justify selection of this
method to measure travel time.

7. Review AccessMod documentation on how to handle roads and water bodies features in
modeling access rather than eliminating them.

8. How did you in-cooperate the changing elevation/slope in walking and cycling
speed? Consider using the Tobbler’s equation.

9. Consider including the odd ratios of the stillbirths and intrapartum stillbirths
in Table 1 and Table 2 respectively.

10. Did you identify the referral cases? From which level and type of facilities were
they referred from?

11. Review on how to capture the SD, OR and CI ranges correctly within a sentence.
Include the quantifying percentages in the sentences rather than at the end.

12. What could be the possible reasons why the majority of women used the Gombe
tertiary hospital rather than their nearest facilities within the state, before you
compare with other state and prior studies?

13. State the specific mean travel times of the prior studies you are comparing with,
state the localities of those studies and state possible reasons for attaining
similar mean travel time in your study.

14. Discuss and put into context why women with stillbirth are poorer, less educated
etc in your study setting. Compare with prior similar studies.

15. Can the lowest administration unit be used for the purpose of uniformity to
represent the residence location of the mother instead of using two (i) areas of
residence and, (ii) town of residence?

16. AccessMod5 travel time surface is normally generated on the assumption that the
care seeker uses the nearest health facility. It’s one of the limitations of the
travel time model. How did you overcome this limitation as stated in your
discussion? (Review on the merits and demerits of using a travel time model).

17. Discuss specific interventions needed for quality obstetric and newborn care in
your recommendation section in view of your findings.

18. Strengthen the conclusion to summarize the study, major findings and way forward
on reducing stillbirth counts in Gombe state.

6. PLOS authors have the option to publish the peer
review history of their article (what does this mean?). If published, this will
include your full peer review and any attached files.

If you choose “no”, your identity will remain anonymous but your review may still be
made public.

**Do you want your identity to be public for this peer review?** For
information about this choice, including consent withdrawal, please see our
Privacy Policy.

Reviewer #1: **Yes: **Jesse Gitaka

---

## [Author Response · Author response to Decision Letter 0]

26 Nov 2020

Journal requirements

 Journal requirements 

1. Please ensure that your manuscript meets PLOS ONE's style requirements, including
those for file naming. The PLOS ONE style templates can be found HERE and HERE

Response: Thanks for providing the guidelines. We have revised the manuscript to meet
the guidelines.

2. Please include additional information regarding the survey or questionnaire used
in the study and ensure that you have provided sufficient details that others could
replicate the analyses. For instance, if you developed a questionnaire as part of
this study and it is not under a copyright more restrictive than CC-BY, please
include a copy, in both the original language and English, as Supporting
Information. Moreover, please include more details on how the questionnaire was
pre-tested, and whether it was validated. 

Response: We have now included a copy of the data collection tool (questionnaire) as
supporting information (S1 File) because we developed the questionnaire ourselves
and it is not under any copyright. We indicate in the manuscript that the
questionnaire was pretested (Page 6, line 2).

3. In your Methods section, please provide additional information about the
participant recruitment method and the demographic details of your participants.
Please ensure you have provided sufficient details to replicate the analyses such
as: a) the recruitment date range (month and year), b) a description of any
inclusion/exclusion criteria that were applied to participant recruitment, c) a
table of relevant demographic details, d) a statement as to whether your sample can
be considered representative of a larger population, e) a description of how
participants were recruited, and f) descriptions of where participants were
recruited and where the research took place. 

Response: The recruitment date range is included on page 5, lines 25-26 of the
manuscript (1st January 2019 to 31st December 2019).

The inclusion criterion was women who experienced stillbirths in our facility between
1st January to 31st December 2019 and who consented to participation in the study
(Page 5, line 26-28). We have now included information on the exclusion criteria on
page 5, lines 29-30).

We have included a statement as to whether our sample can be considered
representative of a larger population (Page 5, line 30 to page 6, line 1-3).

We have provided some description on how participants were recruited (page 6, line
9-14).

Information on where participants were recruited and where the research took place is
provided on page 5, line 25).

4. We suggest you thoroughly copyedit your manuscript for language usage, spelling,
and grammar. If you do not know anyone who can help you do this, you may wish to
consider employing a professional scientific editing service. 

Response: The manuscript has been copyedited by one of the authors (Winfred
Dotse-Gborgbortsi) to improve clarity, grammar and paragraphs.

5 Your ethics statement should only appear in the Methods section of your manuscript.
If your ethics statement is written in any section besides the Methods, please move
it to the Methods section and delete it from any other section. Please ensure that
your ethics statement is included in your manuscript, as the ethics statement
entered into the online submission form will not be published alongside your
manuscript. 

Response: We have now moved the ethics statement to the ethics section of the
manuscript (page 8, line 14-17).

6. We note that Figure 2 in your submission contain map images which may be
copyrighted. All PLOS content is published under the Creative Commons Attribution
License (CC BY 4.0), which means that the manuscript, images, and Supporting
Information files will be freely available online, and any third party is permitted
to access, download, copy, distribute, and use these materials in any way, even
commercially, with proper attribution. For these reasons, we cannot publish
previously copyrighted maps or satellite images created using proprietary data, such
as Google software (Google Maps, Street View, and Earth).

We require you to either (a) present written permission from the copyright holder to
publish these figures specifically under the CC BY 4.0 license, or (b) remove the
figures from your submission 

Response: Updated figures 1, 3 and 4 are maps which were made by the authors. The
updated maps have not been previously published and were produced by the authors
with administrative boundaries data (basemaps) from geoBoundaries (https://journals.plos.org/plosone/article?id=10.1371/journal.pone.0231866),
and OpenStreetMap (https://www.openstreetmap.org/#map=14/10.2251/11.1767&layers=G).
These basemap sources (geoBoundaries and OpenStreetMap) are online, open license
resources of the geographic boundaries of political administrative divisions of
countries.We have included this information in the caption of Figures 1, 3 and
4.

7. Please include captions for your Supporting Information files at the end of your
manuscript, and update any in-text citations to match accordingly. Please see our
Supporting Information guidelines for more information: http://journals.plos.org/plosone/s/supporting-information

Response: We have provided the caption for our supplementary table.

Comments for the Authors (Reviewer #1) 

1. Wariri et al looked at the influence of travel time to health facilities on
stillbirths in Gombe, Nigeria. This is an exciting study in attempt at shedding
light on the myriad of factors that contribute to the unacceptably high rate of
stillbirths in most African countries. However, they need to address the issues
below to improve the quality of the publication. 

Response: We thank the editor and the reviewers for the time spent in reviewing our
manuscript and for providing detailed comments which we believe have improved the
quality of our manuscript.

We have provided a point-by-point response to the reviewers’ comments below.

Note that all references to page and line numbers are based on the revised manuscript
document with tracked changes.

2. What is the influence of existing referral strategy on decision making on where to
seek care first by mothers? Is there a bypassing dynamic in the study population?
What shapes it? 

Response: Thank you for raising these important questions which would further
contextualise the findings reported in this manuscript. While anecdotal reports
exist related to the bypassing of primary and secondary health facilities by mothers
to seek care in the only tertiary health facility in Gombe (our study population),
to the best of our knowledge, there is no documented peer-reviewed evidence to back
this claim. Thus, we are unable to explicitly provide an answer on the influence of
the existing referral strategy on the decision making on where to seek care by
mothers or what shapes such behaviour. 

In an attempt to answer these very important questions, we had tried to explore the
reasons behind bypassing the nearest primary or secondary facilities by the mothers
in our study population (Page 15, paragraph 2). In Page 15, Lines 19-23, we
identified the quality of maternal and newborn services offered public-sector
primary and secondary facilities as possible reasons shaping bypassing or decision
on where to seek care by mothers.

3. It would have been nice to show flow of patients to the tertiary hospital in a map
to better understand the influence of residence on choice of facility. 

Response: Thank you for this brilliant suggestion. We have added Fig 3 which shows
the flow patterns of still births only. We excluded the controls (live births) from
this new map.

4. Geographically inaccessibility directly affects the second delay in the
three-delay concept by Thaddeus and Maine, rather than the indirect impact it has on
the first delay. Discussion on second delay needed as well. 

Response: We agree with the reviewer on the point raised here. In fact, in the
introduction section, on page , lines 6 – 10, we discuss and imply that geographical
inaccessibility indirectly impact the first-delay by creating a lack of interest to
seek care, as well as impacting directly the second-delay by increasing the travel
time in reaching health facilities. We appropriately referenced our assertions (ref
12 – 14).

5. Is there a recommended travel time to nearest hospital facilities which reduces
occurrence of stillbirths during delivery? 

Response: To our knowledge, there is no recommended travel time to nearest health
facility which reduces the occurrence of stillbirths.

6 Include the precise coordinate location of the study setting, inclusion of study
area map will be helpful. 

Response: Thank you for the suggestion. We have added a map of the study area. We did
not add the precise coordinate of the study location but rather shown it on the new
map. See Fig 1 on page 5

7. Why did you select the travel time model via AccessMod? Justify selection of this
method to measure travel time. 

Response: Thank you for this comment. We have now justified why we used AccessMod and
travel time estimates. A summary of our revision is that, travel time is better than
Euclidean and network distance. Also, AccessMod is free, easy to use and widely used
for similar purposes. See page 7, lines 2 – 5.

8. Review AccessMod documentation on how to handle roads and water bodies features in
modelling access rather than eliminating them. 

Response: We have revised this section to improve clarity. We did not eliminate roads
and water bodies. What AccessMod does is to remove the artificial bridge. This is
explained in Section 5.5.2 and other parts of the AccessMod user manual. See lines
15 – 18 on page 7.

9. How did you in-cooperate the changing elevation/slope in walking and cycling
speed? Consider using the Tobbler’s equation. 

Response: The variations in walking/cycling speed due to changing elevation was
implemented using the Tobler’s equation. As we wanted to improve readability for a
wider audience including health workers and policy makers with less or without GIS
knowledge, we avoided using such technical terms. However, we have revised this
section to mention Tobler and direct readers to the user manual and an associated
paper (https://link.springer.com/article/10.1186/1476-072X-7-63) for more
technical readers. See lines 19 – 22 on page 7.

10. Consider including the odd ratios of the stillbirths and intrapartum stillbirths
in Table 1 and Table 2 respectively. 

Response: Thank you for this comment. We, however, retain the proportions and
percentages as presented in Table 1 and 2 in the manuscript. We retain these rather
than the odds ratios as suggested because the tables show the comparison of the
frequency of key sociodemographic characteristics by livebirths vs stillbirths
(Table 1) and antepartum vs intrapartum stillbirths (Table 2). We were NOT comparing
the relative odds of the occurrence of the outcome of interest (stillbirths OR
antepartum/ intrapartum stillbirths), given exposure to the variable of interest (in
this case, the sociodemographic characteristics). Table 4 includes odds ratios
because we were comparing the odds of the outcome (stillbirths) given the exposure
(travel time to nearest facility and to the FTHG).

11. Did you identify the referral cases? From which level and type of facilities were
they referred from? 

Response: We identified the cases that were referred in Table 1 and Table 2. Table
shows that 73% (116/159) of stillbirths (cases) were referred compared to 17%
(27/159) of live births (controls). Table 2 (page 9) gives the comparison of
referral status between antepartum and intrapartum stillbirths.

Both tables also provide information on the level/type of facility (facility referred
from) where the cases or cases have been referred from. For example, in Table 1
(page 8), 35% (40/116) of stillbirths were referred from primary healthcare centres,
47.9% (56/116) from secondary healthcare facilities etc. Again, Table 2 (page 9)
gives the comparison of which facilities referred from (type/level), for antepartum
and intrapartum stillbirths

12. Review on how to capture the SD, OR and CI ranges correctly within a sentence.
Include the quantifying percentages in the sentences rather than at the end. 

Response: Thank you for this comment. We have now implemented the suggested edits
throughout the result section of the manuscript.

13. What could be the possible reasons why the majority of women used the Gombe
tertiary hospital rather than their nearest facilities within the state, before you
compare with other state and prior studies? Thank you for this comment. We have
edited paragraph 2 of the discussion section (page 15) to reflect this comment. In
brief, we argue that the poor quality and variability of public primary and
secondary facilities within the State could partly explain why most women sought
care in the FTH Gombe, rather that their nearest facilities within the state.

14. State the specific mean travel times of the prior studies you are comparing with,
state the localities of those studies and state possible reasons for attaining
similar mean travel time in your study 

Response: We have provided additional information as requested for the studies we
compared our finding with. See page 15, line 24-28. We were unable to compare our
mean travel time to the more related study in Africa by Karra et al (38) because
their travel times are self-reported and prone to reporting and recall bias. More
so, their travel time estimates were presented in grouped categories. 

15. Discuss and put into context why women with stillbirth are poorer, less educated
etc in your study setting. Compare with prior similar studies. 

Response: Thank you for raising this comment. We have now contextualised why women
with stillbirths are poorer and less educated (page 16, lines 1-9). We also compared
our finding with prior similar studies from Gombe (our setting), Nepal, and Span
(page 16, lines 8-9).

16. Can the lowest administration unit be used for the purpose of uniformity to
represent the residence location of the mother instead of using two (i) areas of
residence and, (ii) town of residence? 

Response: Thank you for the suggestion. We have generalised location address to town
of residence. We reported only town level. We have changed the wording in the
methods and clarified that we considered suburbs of the Gombe city itself as towns
else the whole Gombe City as one unit/town will be spatially incomparable in size to
the other towns. Page 6 lines 15 - 19

17. AccessMod5 travel time surface is normally generated on the assumption that the
care seeker uses the nearest health facility. It’s one of the limitations of the
travel time model. How did you overcome this limitation as stated in your
discussion? (Review on the merits and demerits of using a travel time model). Thank
you for reminding us to this inherent AccessMod limitation. 

Response: Actually, this limitation was a strength in our analysis. First, we
modelled travel times to the nearest health facility. Subsequently, we used only the
tertiary facility at Gombe as our destination as all the study participants accessed
care there. Therefore, we modelled a theoretical/expected/ideal travel time to the
nearest health facility and then to the Federal Teaching Hospital Gombe where all
study participants sought care. Thus, we avoided the assumption of modelling
everyone to the nearest facility similar to studies using the Demographic and Health
Survey data due to unknown destinations. We have explained this in our discussion
(page 17, lines 17-23), thus, no revisions were implemented.

18. Discuss specific interventions needed for quality obstetric and newborn care in
your recommendation section in view of your findings. 

Response: We have expanded this section to include specific recommendations needed
for quality obstetric and newborn care (page 17, lines 4-7).

19. Strengthen the conclusion to summarize the study, major findings and way forward
on reducing stillbirth counts in Gombe state. 

Response: We have now strengthened the conclusion section of the manuscript as
suggested (page 18, lines 5-12).

to Reviewers__PONE-D-20-30030.docx
---

## [Decision Letter · Decision Letter 1]

18 Dec 2020

PONE-D-20-30030R1

The influence of travel time to health facilities on stillbirths: a geospatial
case-control analysis of facility-based data in Gombe, Nigeria

PLOS ONE

Dear Dr. Wariri,

Thank you for submitting your manuscript to PLOS ONE. After careful consideration, we
feel that it has merit but does not fully meet PLOS ONE’s publication criteria as it
currently stands. Therefore, we invite you to submit a revised version of the
manuscript that addresses the points raised during the review process.

There is a remaining comment that needs to be addressed.

Please submit your revised manuscript by Feb 01 2021 11:59PM. If you will need more
time than this to complete your revisions, please reply to this message or contact
the journal office at plosone@plos.org. When
you're ready to submit your revision, log on to https://www.editorialmanager.com/pone/ and select the 'Submissions
Needing Revision' folder to locate your manuscript file.

If you would like to make changes to your financial disclosure, please include your
updated statement in your cover letter. Guidelines for resubmitting your figure
files are available below the reviewer comments at the end of this letter.

We look forward to receiving your revised manuscript.

Kind regards,

Frank T. Spradley

Academic Editor

PLOS ONE

Reviewers' comments:

Reviewer's Responses to Questions

**Comments to the Author**

1. If the authors have adequately addressed your comments raised in a previous round
of review and you feel that this manuscript is now acceptable for publication, you
may indicate that here to bypass the “Comments to the Author” section, enter your
conflict of interest statement in the “Confidential to Editor” section, and submit
your "Accept" recommendation.

Reviewer #1: All comments have been addressed

2. Is the manuscript technically sound, and do the data
support the conclusions?

Reviewer #1: Yes

3. Has the statistical analysis been performed
appropriately and rigorously? 

Reviewer #1: Yes

4. Have the authors made all data underlying the
findings in their manuscript fully available?

Reviewer #1: Yes

5. Is the manuscript presented in an intelligible
fashion and written in standard English?

Reviewer #1: Yes

6. Review Comments to the Author

Reviewer #1: The authors have addressed all our comments. However, they need to
clarify how assent was achieved for the participants below consenting age.

7. PLOS authors have the option to publish the peer
review history of their article (what does this mean?). If published, this will
include your full peer review and any attached files.

If you choose “no”, your identity will remain anonymous but your review may still be
made public.

**Do you want your identity to be public for this peer review?** For
information about this choice, including consent withdrawal, please see our
Privacy Policy.

Reviewer #1: **Yes: **Jesse Gitaka

---

## [Author Response · Author response to Decision Letter 1]

19 Dec 2020

COMMENTS: Reviewer #1: The authors have addressed all our comments. However, they
need to clarify how assent was achieved for the participants below consenting
age

RESPONSE: In this study, we considered the participants who were below the age of 18
years (the traditional age of consenting in our setting) as ‘emancipated minors’
because all of them were already married. Thus, consent (rather than assent) was
also sought from these participants in a similar manner to those women who were 18
years older. Note that teenage marriage is common in our study setting. We have
stated this clearly in the manuscript in page 5, lines 27 - 31

---

## [Editor Report · Decision Letter 2]

26 Dec 2020

The influence of travel time to health facilities on stillbirths: a geospatial
case-control analysis of facility-based data in Gombe, Nigeria

PONE-D-20-30030R2

Dear Dr. Wariri,

We’re pleased to inform you that your manuscript has been judged scientifically
suitable for publication and will be formally accepted for publication once it meets
all outstanding technical requirements.

Kind regards,

Frank T. Spradley

Academic Editor

PLOS ONE

---

## [Editor Report · Acceptance letter]

29 Dec 2020

PONE-D-20-30030R2 

The influence of travel time to health facilities on stillbirths: A geospatial
case-control analysis of facility-based data in Gombe, Nigeria. 

Dear Dr. Wariri:

I'm pleased to inform you that your manuscript has been deemed suitable for
publication in PLOS ONE. Congratulations! Your manuscript is now with our production
department. 

Kind regards, 

on behalf of

Dr. Frank T. Spradley 

Academic Editor

PLOS ONE